# The global impact of COVID-19 on tuberculosis: A thematic scoping review, 2020–2023

**Michael H. Marco**[1,2]*, **Sevim Ahmedov**[1], **Kenneth G. Castro**[1,3]

**1** TB Division, Office of Infectious Diseases, Bureau for Global Health, United States Agency for International Development, Washington, District of Columbia, United States of America, **2** Global Health Technical Assistance and Mission Support, Vienna, Virginia, United States of America, **3** Rollins School of Public Health, School of Medicine, Emory/Georgia TB Research Advancement Center, Atlanta, Georgia, United States of America

* mhm2109@gmail.com

## Abstract

### Background

This thematic scoping review of publications sought to understand the global impact of COVID-19 on tuberculosis (TB), interpret the scope of resonating themes, and offer policy recommendations to stimulate TB recovery and future pandemic preparedness.

### Data sources

Publications were captured from three search engines, PubMed, EBSCO, and Google Scholar, and applicable websites written in English from January 1, 2020, to April 30, 2023.

### Study selection

Our scoping review was limited to publications detailing the impact of COVID-19 on TB. Original research, reviews, letters, and editorials describing the deleterious and harmful—yet sometimes positive—impact of COVID-19 (sole exposure) on TB (sole outcome) were included. The objective was to methodically categorize the impacts into themes through a comprehensive review of selected studies to provide significant health policy guidance.

### Data extraction

Two authors independently screened citations and full texts, while the third arbitrated when consensus was not met. All three performed data extraction.

### Data synthesis/Results

Of 1,755 screened publications, 176 (10%) covering 39 countries over 41 months met the inclusion criteria. By independently using a data extraction instrument, the three authors identified ten principal themes from each publication. These themes were later finalized through a consensus decision. The themes encompassed TB's care cascade, patient-centered care, psychosocial issues, and health services: 1) case-finding and notification (n =

**Data Availability Statement:** For our scoping review, we have documented for reproducibility the search terms employed in the three search engines (PubMed, EBSCO, Google Scholar) in S2 Text.

**Funding:** The source of funding (e.g., salary) for the three authors was from the TB Division in the Bureau for Global Health at USAID. The funders played no role in the content. The views and opinions expressed in this paper are those of the authors and not necessarily the views and opinions of USAID. No grants or special funding was obtained.

**Competing interests:** The authors have declared that no competing interests exist.

45; 26%); 2) diagnosis and laboratory systems (n = 19; 10.7%) 3) prevention, treatment, and care (n = 22; 12.2%); 4) telemedicine/telehealth (n = 12; 6.8%); 5) social determinants of health (n = 14; 8%); 6) airborne infection prevention and control (n = 8; 4.6%); 7) health system strengthening (n = 22; 13%); 8) mental health (n = 13; 7.4%); 9) stigma (n = 11; 6.3%); and 10) health education (n = 10; 5.7%).

## Limitations

Heterogeneity of publications within themes.

## Conclusions

We identified ten globally generalizable themes of COVID-19's impact on TB. The impact and lessons learned from the themed analysis propelled us to draft public health policy recommendations to direct evidence-informed guidance that strengthens comprehensive global responses, recovery for TB, and future airborne pandemic preparedness.

## Background

Before the first COVID-19 cases were reported in Wuhan, China, in late 2019 [1, 2] and the World Health Organization's (WHO) declaration of COVID-19 as a global pandemic on March 11, 2020 [3], tuberculosis (TB) was the world's leading cause of death from a single infectious disease, causing 1.4 million deaths in 2019 [4]. Global TB deaths declined by 35% between 2009 and 2019, due in part to Member State commitments to WHO's 2015 "End TB Strategy" [5] and the 2018 United Nations High-Level Meeting declaration to end the TB epidemic [6].

The global spread of COVID-19 likely derailed the promise of and trajectory for ending TB in this decade. COVID-19 wiped out over 12 years of hard-fought gains, such as increased TB detections and decreased TB-related mortality [7]. The initial marked drop in the reported number of newly diagnosed TB cases dropped from a peak of 7.1 million in 2019 to 5.8 million in 2020––a decline of 18% [8]. It is estimated that there will be 4,702,800 TB cases and an additional 1,044,800 TB deaths worldwide between 2020 and 2025 due to the disruptions in TB detection and treatment during lockdowns, and the prioritization of COVID-19 services [9]. By 2022, the WHO [10] stated, "...the COVID-19 pandemic continues to have a damaging impact on access to TB diagnosis and treatment and the burden of TB disease."

We conducted a scoping review to explore the impact, effect, and aftermath of COVID-19 on all aspects of TB, from the clinical cascade to patient-centered care, provision, and psychosocial issues. Based on our analysis, we identified ten principal themes and expanded the traditional model of a scoping review by offering policy recommendations aligned with these principal themes. This review and discussion target policymakers. As COVID-19 continues to cause morbidity and mortality, and the threat of future airborne infection pandemics remains a real possibility, it seemed prudent to offer TB policy recommendations to global and national leaders to build and maintain a robust TB infrastructure with surge capacity for response, recovery, and resilience.

## Methods

The analysis of publications and grey literature was conducted to identify and define resonating themes of COVID-19's global impact on TB and inform policy recommendations to assist

in TB resilience, recovery, and future pandemic preparedness. This scoping review was conducted in accordance with Joanna Briggs Institute (JBI) methodology and protocol template (S1 Text, and published at doi: https://doi.org/10.6084/m9.figshare.24566842.v1) and PRISMA Extension for Scoping Reviews guidelines with a checklist (S2 Table) [11, 12]. Regarding the ethical considerations of using data in a scoping review, our team observes three essential features: 1) researchers must ensure all personal information is kept confidential; 2) sensitive or identifiable personal data should not be used in the publication or presentation of study data; and 3) if the study involves human subjects, informed consent/ethical approval should be obtained.

## Data sources and searches

We included qualitative and quantitative original research, case series, letters to the editor, editorials, and review articles that met the inclusion criteria. On May 1, 2023, publications written in English were captured from three search engines: PubMed, EBSCO, Google Scholar, and global health multilateral websites from January 1, 2020, to April 30, 2023. The PubMed search terms, which originated from the consensus in review team meetings, were:

> *((Mycobacterium tuberculosis or tuberculosis or TB [MeSH Terms])) AND ((COVID-19 or COVID or pandemic COVID-19 or coronavirus or sars-cov-2 [MeSH Terms])) AND ((primary healthcare or health services or healthcare system [MeSH Terms]))).* (S2 Text)

This search strategy was designed to be wide-ranging and purposeful to include as many studies as possible from low- and middle-income counties (LMIC). Identified citations were uploaded to EndNote 20 (Clarivate) and transferred to COVIDENCE systematic review software for screening.

## Study selection

The Population, Concept, and Context (PCC) framework [11] was applied to assist the review team in setting eligibility criteria. The inclusion criteria stipulated that the literature must discuss COVID-19 (the sole exposure) as the impact, effect, or consequence of any aspect of TB (the sole outcome). Letters to the editor and editorials were required to contain original data or substantive recommendations. We excluded 1) published abstracts; 2) magazines; 3) online pre-publications that were never published; and 4) literature focusing on the clinical sequelae of COVID-19 and TB co-infection, or co-infection dynamics between TB and COVID-19 with other illnesses (e.g., HIV, diabetes).

MHM and KGC conducted independent screening of titles and abstracts; when disagreement occurred, SA adjudicated. MHM and KGC conducted independent full-text reviews with SA, confirming ineligible publications.

## Data extraction and quality assessment

The three authors independently extracted data using a data extraction instrument developed by the review team (S1 Table). They used the instrument to interpret each publication's suggested theme and later decided on one by consensus. Descriptive variables extracted included author(s), geography, publication year, publication type, specifics of the original research publications, and perceived themes.

## Data synthesis and analysis

Determining the resonating themes of publications and how many were decided by consensus. Analysis of a publication's sub-themes was essential for proper categorization. While a publication's title/topic pointed to a specific theme (e.g., case-finding and notification), there were instances when it was added to a different theme because of substantial, pertinent discussions on additional underlying causes or solutions. Themes and their respective publications were presented in tabular form (S3 Table) with a narrative summary describing how each related to COVID-19's global impact on TB.

## Role of the funding source

This work was funded by the United States Agency for International Development.

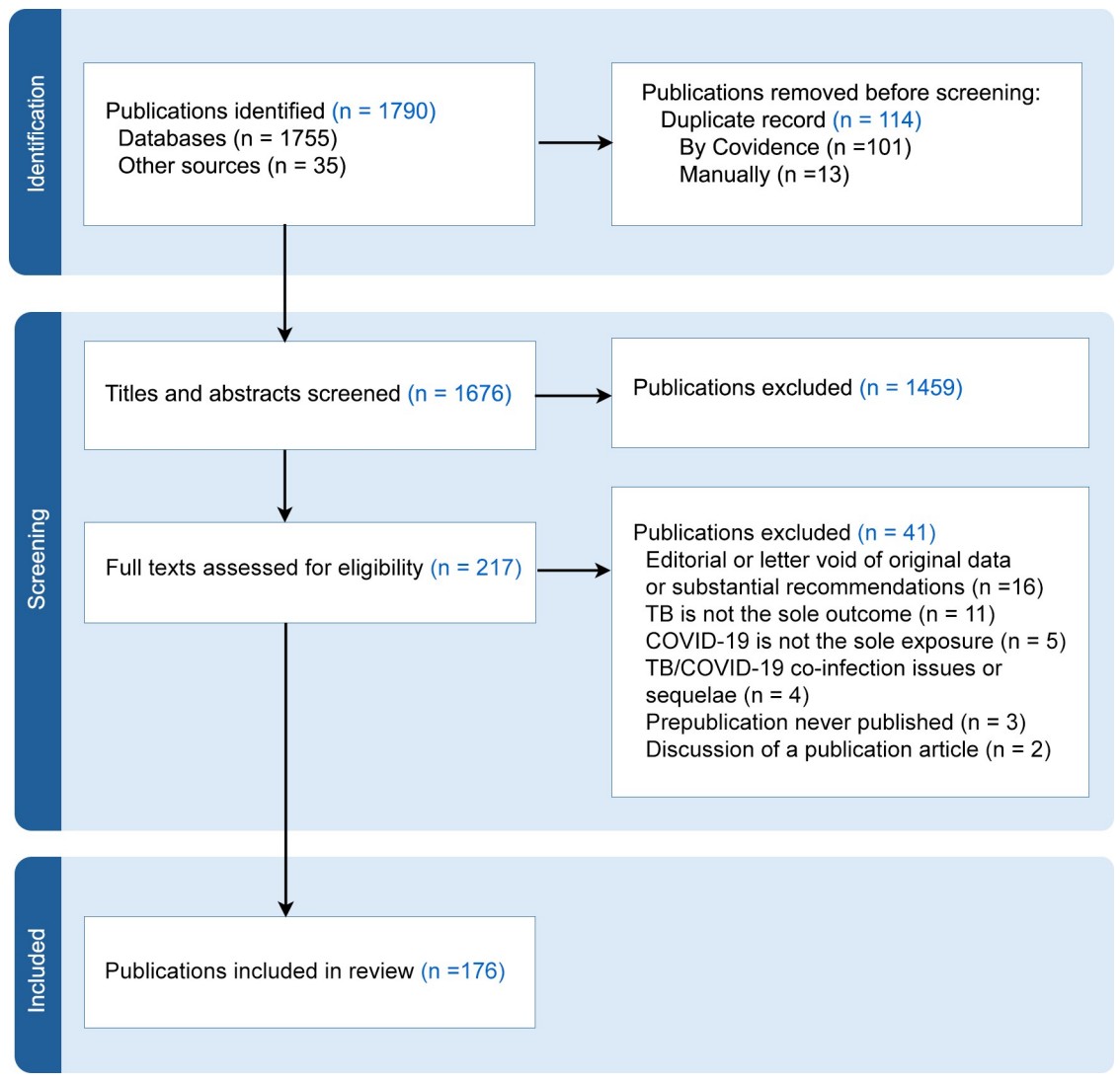

**Fig 1. Scoping review schema.**

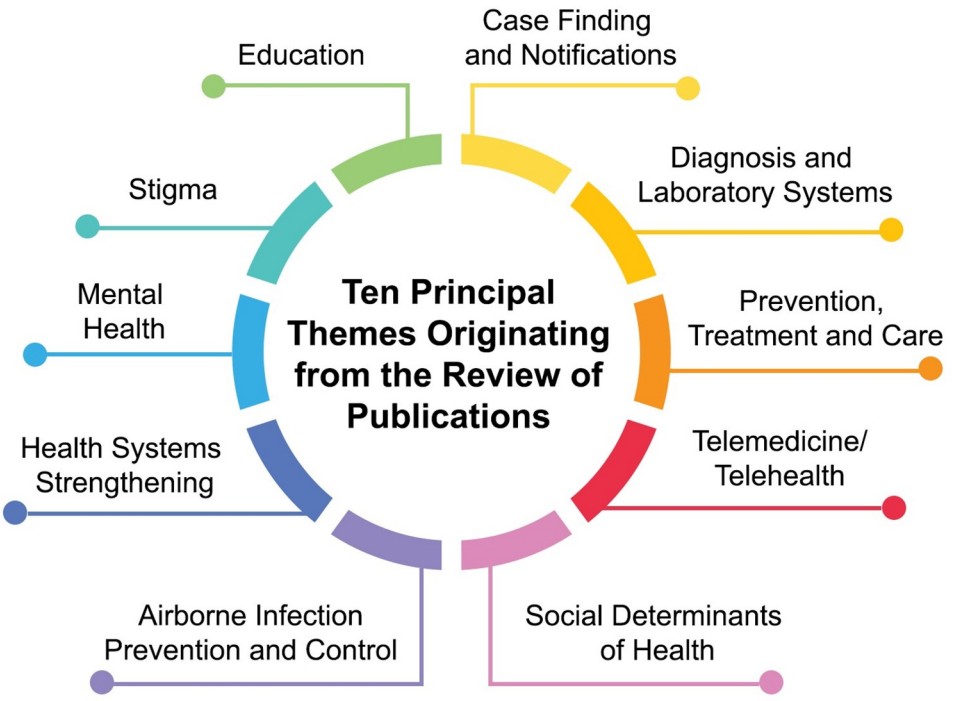

**Fig 2. Ten principal themes characterizing the global impact of COVID-19 on TB.**

## Results

Fig 1 describes the selection process for publication. Three search engines identified 1,755 publications: PubMed (n = 701), EBSCO (n = 105), and Google Scholar (n = 701). Through citation searching, 35 additional publications were captured; 4 were grey literature. There were 114 duplicate titles, leaving 1,674 publications for title and abstract screening. There was agreement from MHM and KGC that 1,426 were irrelevant––i.e., not meeting inclusion criteria–– with a discrepancy of 33. SA adjudicated, characterizing all 33 irrelevant, leaving 1,459 discarded and 217 publications moving to full-text review. Of the full texts reviewed by MHM and KGC, 41 were ineligible, with confirmation by SA. Thus, 176 publications went forward for data capturing and analysis.

In our analyses of 176 publications from 39 countries covering 41 months, ten principal themes (Fig 2) were identified characterizing the breadth of the global impact of COVID-19 on TB: 1) case-finding and notification (n = 45; 26%); 2) diagnosis and laboratory systems (n = 19; 10.7%); 3) prevention, treatment, and care (n = 22; 12.2%); 4) telemedicine/telehealth (n = 12; 6.8%); 5) social determinants of health (n = 14; 8%); 6) airborne infection prevention and control (n = 8; 4.6%); 7) health system strengthening (n = 22; 13%); 8) mental health (n = 13; 7.4%); 9) stigma (n = 11; 6.3%); and 10) health education (n = 10; 5.7%).

### Case-finding and notifications

A quarter of the publications (n = 45) from over 20 counties [7, 9, 13–55] addressed the deleterious impact of COVID-19 on active TB case-finding, notification, and contact tracing during multiple lockdowns and restrictions. Globally, TB case-finding and notification decreased by an estimated 18% between 2019 and 2020 [8], yet resilient TB programs initiated measures to address this drop-off when community health workers (CHWs) were overwhelmed by

COVID-19 testing and response activities. A retrospective data analysis of the impact of COVID-19 on TB case notification and other indicators from Migliori et al. [39] covered five continents and drew data from 43 TB centers in 19 countries. TB case notification decreased from 32,898 in 2019 to 16,396 in 2020; the most precipitous decline occurred in March 2020. Only two countries, Australia, and Singapore, and one state (Virginia, in the U.S.) did not report decreases.

To help blunt the precipitous global decline in TB case notification, finding, and detection, Sahu, et al. [7] stressed the urgency of resilience and recovery from COVID-19 by fast-tracking the 2018 United Nations General Assembly High-Level Meeting TB targets [6] and aligning vaccination services with active early case finding and other community-based TB services.

Publications from several countries described the diversion of resources from TB to COVID-19 and how multiple lockdowns during the pandemic negatively impacted active case-finding and contact investigations, markedly decreasing the detection of people with TB disease or TB infection [7, 15, 17, 26, 30, 43, 52–55]. A few publications [7, 17, 53–55] acknowledged that digital tools developed and deployed for COVID-19 contact tracing could help facilitate future TB contact investigations. Pai, et al. [54] described the need for targeted active case-finding initiatives employing portable digital X-ray systems with artificial intelligence (AI) software. Ruhwald, et al. [53] emphasized that the novel use of COVID-19 molecular technologies and bi-directional testing will benefit TB diagnosis and reduce reliance on suboptimal tools, such as smear microscopy. Sahu, et al. [7] asserted that these advanced diagnostics must be readily implemented in the community to meet people where they live. Finally, Chan, et al. [17] highlighted the benefits and risks of using molecular technology in the community.

## Diagnosis and laboratory systems

Nineteen publications [56–74] focused on COVID-19's detrimental impacts on TB diagnosis, laboratory capacity and systems, and the supply chain.

Maurer, et al. [73] detailed a WHO survey of 31 national TB reference laboratories in the European Union and the United Kingdom, reporting that COVID-19-related disruptions to TB laboratory services peaked from March to June 2020. The core laboratory setbacks were: 1) sample turnaround time; 2) access to external quality assessment; and 3) availability of diagnostic services.

Tovar, et al. [72] conducted a modelling analysis to determine the impact of COVID-19 setbacks in TB diagnostic and laboratory services on patient mortality in India, Indonesia, Kenya, and Pakistan in 2022. They calculated that stark, pandemic-related reductions in new TB diagnoses could result in 378,000 excess deaths across the four countries.

Integrating COVID-19 and TB testing was proposed by MacLean, et al. [63] as a solution to identify and diagnose more people with TB disease. They contended that during the pandemic, too many people refrained from accessing healthcare, including essential TB testing. They provided integrated testing recommendations for 1) urban settings with the highest TB prevalence and those vulnerable to COVID-19; 2) rural settings in high-burden countries to improve the quality of patient care; and 3) countries with high HIV prevalence.

In 2020, Awasthi and Singh [74] provided solutions for India's TB diagnostic and laboratory obstacles. They called for the Indian government to increase testing by arranging additional machines and increasing the number of laboratory shifts. Furthermore, they called for the Government's TB Program to offer upfront TB testing with GeneXpert (Cepheid, Sunnyvale, CA) or Truenat (Bigtec Labs, Bangalore, India) instead of smear microscopy.

## Prevention, treatment, and care

Twenty-two publications [75–96] centered on the pandemic's negative impact on TB prevention, treatment, and care in diverse patient populations; some proposed measures to optimize TB treatment.

In a 2023 rapid review from South Korea, Jeong et al. [83] reported that global detection and treatment of TB infection were among the most negatively impacted steps in the TB cascade across low-, middle-, and high-income countries (LMHC). Two retrospective data analyses comparing TB preventive treatment initiation before and during the pandemic reported a 44.7% decline in Addis, Ethiopia [76] and 30% and 66% in Montreal and Toronto, Canada [77], respectively.

The pandemic resulted in global stockouts of BCG vaccine for infants [89, 94]. Namkoong, et al. [89] hypothesized that the vaccine shortage was partly caused by WHO's supplier withdrawal due to production issues, which left UNICEF [97] with a 30% decrease in supply. Another explanation was the renewed interest in basic science research [98] that supported BCG's enhancement of immune responses. This led to off-label use, even after the publication of results from a failed efficacy trial of BCG for COVID-19 prevention [99].

Arega et al. [76] conducted a retrospective data analysis of TB treatment outcomes in Ethiopia, finding that the TB treatment success rate decreased by 17% between March 2019 and March 2020 and that rifampicin resistance (RR) increased by 27.7%.

The increase of RR/multidrug-resistant (MDR) TB was observed in different settings during the pandemic. From the proceedings of the National Academies of Sciences workshop, "Innovations for Tackling Tuberculosis in the Time of COVID-19," Salmaan Keshavjee [90] underscored the necessity of continuing TB treatment during the pandemic by administering the most tolerable and shortest DS-TB, and all-oral DR TB regimens, and promoting treatment adherence.

## Telemedicine/Telehealth

Twelve publications [100–111] discussed how LMIC swiftly implemented telehealth interventions to offset COVID-19 disruptions in TB services. Various digital tools enabled virtual case identification, TB care, treatment, and adherence [4].

A rapid assessment of telemedicine's potential to optimize TB care and treatment was conducted by Klinton et al. [102] during March–December 2020 in seven high-burden countries. The interventions included telemedicine/telehealth platforms (e.g., text messages, phone, and video) enabling consultations, video-observed therapy (VOT) for adherence, refill reminders, and novel diagnostic platforms such as AI-based and portable digital chest X-rays. Researchers noted that digital tools could strengthen the diagnostic capacity of TB programs, enhance patient-centered care, decentralize TB services, and contribute towards progress in achieving the EndTB goals.

Calnan, et al. [107] described a phone-based TB case-finding and case-management intervention launched in two regions of the Philippines between October 2020 and September 2021. Fourteen TB contact centers conducted TB screening and contact investigation and provided information about testing, delivery of test results, and adherence support. Call centers identified 9.2% of people with TB in the region, of which 43.5% (827/1,901) initiated treatment. A cost-benefit analysis compared the new telehealth service versus standard-of-care (SOC) case-finding interventions found that the 1-year cost for implementing call centers totaled USD 557 per patient, only USD 7.00 more than SOC.

A 197-person study by Visca, et al. [108] compared the effectiveness and cost of VOT versus clinic-based directly observed therapy (DOT) in Moldova. They found that VOT significantly

improved adherence (1.29 missed days versus 5.24 missed days) over two weeks spent on clinic transport among VOT patients by 58 hours.

## Social determinants of health

Fourteen publications [112–125] addressed social determinants of health (SDH) and COVID-19's outsized effect on vulnerable populations and offered solutions to mitigate them. Singh et al. [112] summarized the contribution of SDH to an additional 700,000 undiagnosed and missing active TB cases in Southeast Asia between 2019 and 2020, noting that the pandemic adversely impacted the nutritional status and BMI of the affected populations, which, in turn, was likely to have contributed to secondary immunodeficiency and an influx of undiagnosed TB cases.

In a 2020 editorial, Saunders et al. [115] proposed social protection interventions–"safety nets"––for vulnerable populations, noting that impoverished, at-risk TB populations could not work and access funds to provide social protection during COVID-19-related lockdowns. They recommended special provisions, including cash transfers or food parcels for TB-affected households. Notably, they called for psychosocial support for affected households and access to digital technology to improve equity and access to virtual care during lockdowns.

In Brazil, Souza, et al. [123] surveyed healthcare workers about the surge in MDR TB during the first two waves of COVID-19, documenting an association between MDR TB cases and SDH (e.g., poverty, vulnerability, and social risk). Weakened adherence was associated with cuts in social protection and benefits.

## Airborne infection prevention and control

Eight publications [126–133] addressed airborne infection prevention and control and personal protective equipment (PPE) for healthcare workers and people with TB.

Mannan et al. [126] from Joint Effort for Elimination of TB (JEET) in India, a nationwide Global Fund project across 406 districts in 23 states, surveyed 21,750 physicians between February and March 2021. Many survey questions examined infection prevention and control measures (IPC) implemented in their clinics due to COVID-19. Approximately 82% of surveyed providers employed social distancing and increased the interval between patients' appointments. While 70% reported knowledge that IPC measures could decrease TB transmission, 62% initiated PPE use, and only 13% physically implemented physical changes (e.g., air filters and isolation of patient areas).

To prevent airborne infections in TB diagnostic and treatment centers, Awan, et al. [127] proposed several measures at clinics: building and maintaining sizeable outdoor patient waiting areas, installing ventilation and air circulation exhaust systems in clinics, ultraviolet germicidal irradiation lights, and designated sputum expectoration areas.

The shortage of PPE during COVID-19 was echoed in multiple publications from various countries [129–131]. Jain and colleagues [130] reviewed literature published in India in May 2020, which underscored complaints about the paucity of PPE, noting that this made it impossible for HCWs to provide safe, regular healthcare services for people with TB. Moreover, healthcare workers were fearful and reluctant to take samples from people with TB due to a lack of appropriate PPE.

## Health system strengthening

National TB Programs (NTP) in many LMICs were hobbled and underperformed well before the advent of COVID-19. The pandemic imposed an unprecedented burden, leaving programs

and services faltering. Twenty publications examined the need for health system strengthening (HSS) [134–155].

Khan, et al. [139] cited NTP data on widespread disruptions across the care cascade in Pakistan during 2020 due to limited and dwindling resources. Between March 1 and June 30, 2020, GeneXpert machines and hospital isolation wards were re-directed from TB to COVID-19, delayed TB diagnoses. Meanwhile, outpatient TB visits dropped by 59.6%, and hospitalizations declined by 50.7%, increasing the risk of household transmission. Due to staffing shortages, TB treatment was often interrupted, and follow-up visits were delayed or canceled.

Some publications addressed TB differentiated service delivery (DSD) as an essential facet of strengthening health systems [148, 150, 153, 154]. DSD models (such as multi-month dispensing, pickup points for or home delivery of medications, and VOT) quickly emerged, allowing TB services to continue outside the clinic during lockdowns. While piecemeal DSD models for TB were rolled out in the past, these were developed and deployed in an expansive, global, warp-speed offensive during the COVID-19 pandemic [101, 104, 146].

Klinton et al. [142] observed a silver lining for TB HSS, noting global resilience, recovery, and innovative improvements despite the erosion of TB services during the pandemic. The private sector enabled the rapid deployment of innovations that improved TB services and health systems, demonstrating resilience by adapting guidelines, policies, and digital tools to improve accessibility, acceptability, and quality of TB prevention, care, and treatment. The pandemic experience highlighted the importance of strengthening and adapting TB health system services, and the essential role of public-private partnerships in maintaining them. These observed benefits strengthened overall health systems well beyond TB services.

## Mental health

Depression is more prevalent among people with TB than in the general population. A 2020 systematic review and meta-analysis by Ruiz-Grosso, et al. [156] documented a strong association between TB and depression that caused adverse TB treatment outcomes, including poor adherence, loss-to-follow-up, and death. Thirteen publications [157–169] discussed mental health concerns, emphasizing the escalation of fear and anxiety among people with TB during the pandemic.

Loveday, et al. [157] noted that lockdowns and other COVID-19 restrictions in South Africa posed harsh financial consequences for people who were already economically vulnerable, contributing to anxiety, stress, and depression among people with TB.

Pronounced fear and anxiety were evident in an 842-person global survey [162] coordinated by nine TB non-governmental organizations (NGOs). Qualitative and quantitative data were collected from people with TB and survivors, healthcare workers, NTP staff, civil society, and advocates between May 26, 2020, and July 2, 2020. Over half of the people with TB reported feeling increased vulnerability to and fear of contracting COVID-19, which prevented some of them from seeking treatment. This study and others [154–157] mentioned that healthcare workers feared seeing patients because of anxiety about acquiring COVID-19 and transmitting it to their families.

During the pandemic, a novel telehealth approach to assess mental health among people with TB and healthcare providers was implemented in Pakistan [159]. Through regular phone calls, mental health providers checked emotional well-being and screened people for depression and anxiety. Additional support was provided for those who acquired COVID-19.

## Stigma

Eleven publications [170–180] addressed TB-related stigma, the populations most affected, and its adverse effects––especially during COVID-19––and solutions to combat it.

Dheda et al. [180] shared their frank views of TB-related stigma in a 2022 commentary. They asserted that "social stigma kills," noting that it is repugnant for people who battle airborne infectious diseases in isolation to fight social stigma simultaneously.

Mahnoor Islam [173] wrote that TB-related stigma during the height pandemic led to poor treatment outcomes. He asserted that the stigma of respiratory symptoms faced by people with TB, which COVID-19 often compounded, led to a reluctance to visit healthcare facilities, thus enabling drug resistance because of incomplete TB therapy.

Anti-stigma interventions must be specifically tailored to patient groups (e.g., women and girls) in different countries. In a commentary on COVID-19 and TB in Pakistan, Fatima et al. [172] asserted that the best way to alleviate the stigma of TB for women and girls is to develop and employ TB interventions in rural communities with the aid and expertise of lady health workers.

## Health education

Ten publications [181–190] addressed the need for TB patient education programs and community information campaigns to assuage fear, anxiety, and stigmatization, which leads those with TB––and undiagnosed people––to avoid clinics for fear of COVID-19. Likewise, educating TB healthcare workers about COVID-19 and how to reduce their risk is essential for them to feel safe and continue working.

An Iranian qualitative study by Shahnavazi, et al. [190] discussed the reduction in time HCWs had for patient education and follow-up instructions during the pandemic. HCW disclosed that they only had enough time to dispense TB medication before abruptly leaving to attend to COVID-19 duties.

In 2021, Nkereuwem, et al. [181] surveyed European and West African healthcare workers in 2021. Many of the respondents underscored the importance of intensified health education. One respondent said TB education campaigns can help remind the world "not to forget TB" and make people aware that a chronic cough may not be from COVID-19. Another respondent decreed that there must be more public education on TB because COVID-19 overshadowed messaging about other diseases. Some urged NTPs to offer timely statements and guidance on routine TB screening, diagnosis, and treatment during COVID-19 and future pandemics.

Chapman et al. [182] noted that digital health interventions exist to transform TB care by disseminating essential health information that supports treatment adherence and encourages health-seeking behaviors among people with TB. They addressed the need for public health communication campaigns to combat the "infodemic," which they defined as "the rapid spread of false information on TB and COVID-19."

## Policy recommendations from the ten principal themes in the scoping review

**1. Strengthen case-finding linked to care and prevention.** Strengthen community-level and primary healthcare (PHC) active case-finding activities before and during pandemics by integrating digital solutions, such as mHealth platforms, and scale-up the use of screening and diagnostic tools (e.g., deploy ultra-portable digital chest radiographs (CXR) with artificial intelligence (AI)/ computer-assisted diagnosis (CAD) and increased access to rapid molecular diagnostics for TB and other airborne infections with robust linkages to treatment and care.

**2. Advance rapid diagnosis and laboratory systems.** Develop systems to advance all aspects of shared diagnostic platforms, including collecting and transporting all clinical specimens and scheduling to maximize utilization of instruments and enable prompt and accurate identification and transmission of results. Support the use of WHO recommended rapid tests with prompt turnaround time at locations convenient to clients to increase the accuracy of results and decrease time to treatment initiation with appropriate regimens.

**3. Improve outcomes through prevention, treatment, and care.** Improve health outcomes for individuals with TB through:

- Rapid introduction and expanded use of shorter, effective treatment regimens for drug-susceptible and drug-resistant TB

- Person-centered and supportive care and treatment that includes convenient access to services, mHealth digital solutions on apps, multi-month dispensing of medicines, when applicable, and the availability of telemedicine for remote consultation and adherence support

- Referrals for other airborne infectious diseases with available diagnostic and curative services and to healthcare services addressing conditions impacting TB infection and disease progression (e.g., smoking, diabetes, HIV, undernutrition, and alcohol use)

**4. Scale-up telemedicine for adherence and care.** Telemedicine, or the delivery of healthcare services at a distance with information and communications technology has significant potential to overcome deficiencies of direct healthcare delivery.

- Leverage the newly introduced telemedicine for remote consultations and for improving patient adherence, as well as monitoring for medicines' potential side effects, along with online and offline mobile technology as a means of preventing treatment relapse and the subsequent development or amplification of drug resistance, even outside pandemic situations

- Introduce and scale up Video Directly Observed Therapy (V-DOT) programs as part of comprehensive and patient-centered 'case management' approaches. Develop specific guidelines and standard operating procedures (SOPs), provide training, and monitor implementation

**5. Develop safety nets to address social determinants of health.** Address social determinants of health that negatively affect TB and other airborne diseases by developing safety nets and social support schemes to ensure that individuals living and working in "informal" sectors are protected from the economic consequences of pandemic-related restrictive measures and barriers to accessing diagnostic and treatment services.

**6. Prevent airborne infection by utilizing prevention and control measures.** Build new or retrofit existing facilities that evaluate/treat airborne infectious diseases, including TB diagnostic and treatment centers, to achieve airborne infection control standards related to ventilation, air circulation, the use of ultraviolet germicidal irradiation lights, designated areas for sputum expectoration, and commit to the availability and supply of high-quality personal protective equipment (PPE) for healthcare workers and clients.

**7. Ensure the development of integrated and mutually reinforcing health systems and TB platforms.** While utilizing existing TB platforms (e.g., HR, hospital infrastructure, surveillance systems, PSM) for rapid response to potential airborne pandemics, ensure the development of solid safeguards for simultaneous uninterrupted delivery of TB services. Moreover, health system strengthening (HSS) efforts should consider the need for more plausible and

direct attribution to the improvement of disease-specific outcomes, and vice versa, disease-specific investments should also help build more resilient health systems such as integrating epidemiological and genomic surveillance for developing a comprehensive public health response strategy that is tailored to local needs.

**8. Integrate mental health into people-centered TB prevention and care.** TB care and treatment are complex, and the impact of mental health factors on its success has been historically overlooked. However, the emerging evidence on mental health and TB shows a strong correlation between the prevalence of depression and other mental health issues among TB patients and their adverse effects on their health and survival.

- Integrate mental health care into the entire continuum of TB prevention and care approaches, from screenings and care for depression and anxiety of people with presumptive TB to continuous monitoring of mental care issues during and post-treatment and care for TB patients

**9. Prevent TB stigma in all affected people.** Alleviate stigma among all vulnerable populations in urban and rural areas associated with TB or presenting with respiratory symptoms during airborne pandemics. Measures to address such stigma include:

- Education campaigns with the utilization of local community members who understand local customs and ideas about illness to help deliver interventions for TB and other airborne infections

- Communication campaigns aimed at people with TB or respiratory illnesses to raise awareness about diagnostic, treatment, and supportive services. Campaigns should also provide guidance for the safe return to healthcare clinics during the implementation of non-pharmaceutical measures to prevent COVID-19, influenza, and other airborne infectious illnesses

**10. Garner public awareness through communication and health education.** Intensify public awareness and education about TB and other airborne infectious diseases (e.g., COVID-19, Influenza, Pneumococcal pneumonia) in high-risk areas for individuals at-risk for or diagnosed with TB by relying on culturally appropriate trained community educators creating treatment literacy programs that focus on how to limit transmission, the benefits of early diagnosis, prevention through vaccination, treatment, and adhering to and completing their treatment during COVID-19 and future pandemics.

The ten thematic policy recommendations are designed to inform and stimulate response, recovery, and resilience for various aspects of TB globally. They target a wide range of stakeholders, including those in the global health policy arena, international health ministries, international donor agencies, private sector health providers, and global health multilateral organizations. These recommendations were also based on anticipating a future airborne respiratory pandemic, highlighting the necessity of preparedness on multiple fronts.

While some recommendations (e.g., strengthening case-finding or improving TB outcomes) are not new, they remind us that enhanced efforts to follow best practices for TB control and elimination are integral to meeting END TB goals. Other recommendations, superficially involving vaccines and enhanced diagnostic technology arising from COVID-19, are novel and yet to be fully realized. This will require increased global awareness, commitment, and funding.

## Discussion

This comprehensive scoping review assessed the global impact of COVID-19 on TB. It covers the most prolonged publication period (41 months [January 1, 2000 –April 30, 2023]) and the

most significant number of publications analyzed (n = 176) from 39 countries. The timespan includes publications with data and perspectives from multiple pandemic waves of COVID-19––the SARS-CoV-2 Wuhan-Hu-1, Delta, and Omicron variants.

Scoping or rapid reviews have been published on the impact of COVID-19 on TB. However, these reviews have generally focused on a single issue: infection control [182], primary healthcare services [137], or prevention [83]. This limitation motivated us to conduct a comprehensive scoping review addressing multiple themes, including the impact on TB's cascade of care, psychosocial issues, healthcare services, and their interrelationships.

This scoping review sought to include publications from LMIC to supplement prior publications from high-income countries. By casting a wide net, we were able to detail the diversity of complex TB issues and conditions faced globally by many countries, cultures, and patient populations during the pandemic. The analysis of 176 publications yielded ten principal themes covering TB case-finding, diagnostics, prevention, treatment, infection control, health services, mental health issues (e.g., fear and anxiety), stigma, and community education. Many publications, mainly letters, editorials, and reviews, had overlapping themes.

Several publications, across all themes, offered recommendations for combatting a wide range of TB issues [7, 54, 74, 83, 90, 108, 117, 127, 137, 139, 142, 153, 157, 172, 174, 180, 181]. Some recommendations took a "glass half-full" approach by addressing the potential for positive, post-pandemic impacts on TB––leveraging and retrofitting aspects of the pandemic response, such as new developments in case-finding, diagnostics, and health education.

There are limitations in our scoping review. Some letters to the editors, which were included due to an a priori requirement that original data be discussed, risked the inclusion of data not validated for accuracy. We also detected heterogeneity of publications within themes.

This all-inclusive scoping review established that, despite geographic and income-level heterogeneity, there are several generalizable themes between countries on the impact of COVID-19 on TB. The intensity and magnitude of illness and death from COVID-19 in 2020 rapidly overwhelmed TB programs. Consequently, the TB infrastructure and workforce were repurposed for the COVID-19 response. It will take time for TB programmatic aspects to recover and regain a sense of normalcy and optimal productivity.

Our scoping review aligns with the themes discussed at the most recent 2023 United Nations General Assembly High-Level Meeting (UNHLM) on TB, including commitments to universal health coverage, pandemic prevention, preparedness, and response, and the fight against TB, underscoring the crucial need to simultaneously strengthen and address urgent realities and serious threats [191, 192]. Not surprisingly, several of these themes are echoed in the 2023 "UNHLM on Pandemic Prevention, Preparedness and Response [193]."

## Conclusion

This review demonstrates that the COVID-19 pandemic severely disrupted numerous aspects of TB management, including case finding, clinical care, health services, and psychosocial support. These adverse effects were consistently observed across 39 countries over three years, irrespective of income status. The ten themes identified through this scoping review provide a framework for comprehensive progress by TB programs to simultaneously achieve the End TB goal and address future airborne disease pandemics.

## Supporting information

**S1 Text. Scoping review protocol.**
(DOCX)

**S2 Text. Search terms for citation selection.**
(DOCX)

**S1 Table. Data extraction instrument.**
(DOCX)

**S2 Table. PRISMA ScR checklist.**
(DOCX)

**S3 Table. Thematic categorization of publication.**
(DOCX)

## Acknowledgments

We thank Cheri Vincent, Tara Ornstein, Amy Bloom, and YaDiul Mukadi for discussions regarding the scope of the review and assistance in developing the search string; Tracy Swan for editorial assistance; Margie Joyce and David Pieribone for graphic assistance with Figs 1 and 2; Marcus Renick for graphic assistance with the appendices; and COVIDENCE for offering complimentary software use.

## Author Contributions

**Conceptualization:** Michael H. Marco, Kenneth G. Castro.

**Data curation:** Michael H. Marco, Sevim Ahmedov, Kenneth G. Castro.

**Formal analysis:** Michael H. Marco, Sevim Ahmedov, Kenneth G. Castro.

**Methodology:** Michael H. Marco, Sevim Ahmedov, Kenneth G. Castro.

**Project administration:** Michael H. Marco.

**Supervision:** Kenneth G. Castro.

**Visualization:** Michael H. Marco.

**Writing – original draft:** Michael H. Marco.

**Writing – review & editing:** Sevim Ahmedov, Kenneth G. Castro.

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
