## [Decision Letter · Decision Letter 0]

1 Apr 2024

PGPH-D-24-00435

The Global Impact of COVID-19 on Tuberculosis: A Thematic Scoping Review, 2020-2023

Dear Dr. Marco,

Thank you for submitting your manuscript to PLOS Global Public Health. After careful consideration, we feel that it has merit but does not fully meet PLOS Global Public Health’s publication criteria as it currently stands. Therefore, we invite you to submit a revised version of the manuscript that addresses the points raised during the review process.

This manuscript is a well written one which gives valuable insights  regarding the care of patients with Tuberculosis during the Pandemic

You may please  highlight the ethical considerations regarding usage of data in scoping reviewPlease assure the quality and clarity of the tables and pictures.The recommendations provided can help to develop TB care strategies in the emergence of another pandemic or a public health emergen

We look forward to receiving your revised manuscript.

Kind regards,

Suma  Krishnasastry, MBBS, MD,DNB

Academic Editor

Journal Requirements:

2. Please send a completed 'Competing Interests' statement, including any COIs declared by your co-authors. If you have no competing interests to declare, please state "The authors have declared that no competing interests exist". Otherwise please declare all competing interests beginning with the statement "I have read the journal's policy and the authors of this manuscript have the following competing interests:"

3. Please ensure that Funding Information and Financial Disclosure Statement are matched.

4. In the Funding Information you indicated that no funding was received. Please revise the Funding Information field to reflect funding received.

5. We do not publish any copyright or trademark symbols that usually accompany proprietary names, eg  ©, ®, ™  (e.g. next to drug or reagent names). Please remove all instances of trademark/copyright symbols throughout the text, including  ® & ™ on pages 10 & 15.

6. Please upload a copy of Figures 1 & 2 which you refer to in your text on pages 7, 8 & 21. Or, if the figures are no longer to be included as part of the submission please remove all reference to it within the text.

Additional Editor Comments (if provided):

Reviewers' comments:

Reviewer's Responses to Questions

**Comments to the Author**

1. Does this manuscript meet PLOS Global Public Health’s publication criteria? Is the manuscript technically sound, and do the data support the conclusions? The manuscript must describe methodologically and ethically rigorous research with conclusions that are appropriately drawn based on the data presented.

Reviewer #1: Yes

Reviewer #2: Yes

2. Has the statistical analysis been performed appropriately and rigorously?

Reviewer #1: N/A

Reviewer #2: I don't know

3. Have the authors made all data underlying the findings in their manuscript fully available (please refer to the Data Availability Statement at the start of the manuscript PDF file)?

Reviewer #1: Yes

Reviewer #2: Yes

4. Is the manuscript presented in an intelligible fashion and written in standard English?

Reviewer #1: Yes

Reviewer #2: Yes

5. Review Comments to the Author

Reviewer #1: Peer Review Feedback Report for Manuscript: "The Global Impact of COVID-19 on Tuberculosis: A Thematic Scoping Review, 2020-2023"

Manuscript Number: PGPH-D-24-00435

General Comments:

The manuscript presents a thematic scoping review to understand the global impact of COVID-19 on tuberculosis (TB) care and policy. It methodically categorizes the impacts into themes through a comprehensive review of selected studies, offering valuable insights into TB care during the pandemic and future preparedness. The topic is highly relevant and timely, considering the ongoing global health challenges posed by COVID-19 and TB. The authors have made a commendable effort to synthesize the available literature, and the manuscript aligns well with the PLOS Global Public Health scope.

Specific Comments:

1. Originality and Contribution to the Field:

• The manuscript successfully addresses a significant and timely public health issue. The thematic analysis approach provides a structured literature synthesis, contributing valuable insights. However, it would be beneficial to more explicitly highlight the novel contributions of this review in the introduction and conclusion sections, particularly about existing literature reviews on the topic.

2. Methodology:

• The methodology section is well-structured, detailing the search strategy and selection criteria. However, the manuscript would benefit from a more detailed explanation of the thematic analysis process. Specifically, clarifying how themes were derived and refined during the review process could enhance the transparency and replicability of the study.

3. Statistical Analysis;

• This was not applicable as the exploratory scoping review did not need a meta-analysis.

4. Results and Thematic Analysis:

• The identification of ten principal themes is a major strength of this manuscript. Each theme is relevant and captures a critical aspect of the impact of COVID-19 on TB. Including illustrative quotes or examples from the literature is recommended to enrich the narrative and provide direct evidence for each theme.

• Consider presenting a summary table that maps the identified themes against the number of studies contributing to each theme, geographical distribution, and study design. This would give readers a quick overview of the evidence base supporting each theme.

5. Discussion and Recommendations:

• The discussion provides a thoughtful interpretation of the findings within the broader TB care and policy context. The recommendations offered are pragmatic and relevant. To strengthen this section, the authors could consider discussing potential barriers and facilitators to implementing these recommendations in diverse health system contexts.

• It would be valuable to discuss the limitations of the scoping review methodology, such as the potential for publication bias and the exclusion of non-English language publications.

6. Clarity, Language, and Structure:

• The manuscript is generally well-written and structured.

7. Ethics and Research Integrity:

• The manuscript adheres to high standards of research integrity. The authors have transparently reported their methodology and provided a comprehensive list of articles reviewed. A statement on the ethical considerations of using published literature in scoping reviews could further strengthen this aspect.

8. Reporting Guidelines and Data Availability:

• The authors conducted the review in line with PRISMA-Scr. A PRISMA-ScR checklist could be included as supplementary material.

• A detailed supplementary section was included with the submission.

Conclusion:

This scoping review makes a valuable contribution to understanding the multifaceted impact of COVID-19 on TB care and policy. With minor revisions and enhancements, particularly in clarifying the thematic analysis process and enriching the discussion of findings, this manuscript will be a significant addition to the literature. The recommendations provided can inform future TB care strategies in the context of global health emergencies.

Reviewer #2: My dear authors ;

Many thanks for this good work and it is well written and informative

1- BUT At data sources and researches ; authors use few Mesh term such as not remembered another terms of covid-19 such as 2019 nCOV disease , coronavirus disease 2019, 2019 nCOV infection , Coronavirus disease 2019,2019 novel coronavirus disease as the term covid-19 is considered late by WHO as at onset of pandemic there is many terms not standardized with global term

and also i noticed also you remembered at section of methods your objective of the study is The analysis of publications and grey literature was conducted to identify and define resonating

105 themes of COVID-19’s global impact on TB and inform policy recommendations to assist in TB

106 resilience, recovery, and future pandemic preparedness. but you used limited MESH terms and not remember at your research terms regarding extra pulmonary tuberculosis or preventive measure like missed cases of BCG vaccines as at lock-down there are missed births or neonates not vaccinated or late at intake of BCG vaccine and also missed notification at diagnosis

2- pictures at manuscript are unclear and unfortunately not able to check or review it

6. PLOS authors have the option to publish the peer review history of their article (what does this mean?). If published, this will include your full peer review and any attached files.

**Do you want your identity to be public for this peer review?** For information about this choice, including consent withdrawal, please see our Privacy Policy.

Reviewer #1: **Yes: **Peter Babigumira Ahabwe

Reviewer #2: **Yes: **amr ahmed

---

## [Editor Report · Decision Letter 1]

30 May 2024

The Global Impact of COVID-19 on Tuberculosis: A Thematic Scoping Review, 2020-2023

PGPH-D-24-00435R1

Dear Dr. Marco,

We are pleased to inform you that your manuscript 'The Global Impact of COVID-19 on Tuberculosis: A Thematic Scoping Review, 2020-2023' has been provisionally accepted for publication in PLOS Global Public Health.

Best regards,

Suma  Krishnasastry, MBBS, MD,DNB

Academic Editor